# Flame Retardant Properties and Thermal Decomposition Kinetics of Wood Treated with Boric Acid Modified Silica Sol

**DOI:** 10.3390/ma13204478

**Published:** 2020-10-09

**Authors:** Qiangqiang Liu, Yubo Chai, Lin Ni, Wenhua Lyu

**Affiliations:** Research Institute of Wood Industry, Chinese Academy of Forestry, Beijing 100091, China; liuqq_1353@163.com (Q.L.); chaiyubo@caf.ac.cn (Y.C.); lwhlily@caf.ac.cn (W.L.)

**Keywords:** poplar wood, silica sol, boric acid, cone calorimetric, decomposition kinetics

## Abstract

This paper presents experimental research on the flame-retardant properties and thermal decomposition kinetics of wood treated by boric-acid-modified silica sol. The poplar wood was impregnated with pure silica sol and boric-acid-modified silica sol. The results showed that modifiers can be observed in the cell wall and cell lumen. The ignition time, second peak of the heat release rate, total heat release, and mass loss of the W-Si/B were delayed obviously. The composite silicon modification had a positive impact on carbonization. Thermogravimetric analysis showed that the residual mass of W-Si/B was enhanced and the thermal degradation rate was considerably decreased. By thermal decomposition kinetics analysis, the boric acid can catalyze the thermal degradation and carbonization of poplar wood. In other words, wood treated with boric-acid-modified silica sol showed significant improvement in terms of flame retardancy, compared with wood treated with common silica sol.

## 1. Introduction

Wood is a green environmental protection material, and a natural and renewable resource. With the increasing consumption of wood, the global supply of natural forest wood is increasingly depleted, and it is imperative to enhance the efficient utilization of fast-growing plantation wood especially in China [1]. Compared with natural wood, plantation wood exhibits lower density and strength, so it is also more prone to fire risk [2]. Wood modification is an effective strategy to improve the material properties and qualities of fast-growing plantation wood, and thus expand its applications. Conventional treatment methods such as acetylation [3], heat treatment [4], and impregnation treatment [5,6] can effectively improve the comprehensive performance of wood. Wood is a material with hierarchical porous structure, which allows for the penetration of modifiers and the formation of chemical cross-linking. Petrochemical-derived agents such as low-molecular-weight urea-formaldehyde resins, phenolic resins and acid anhydrides can improve the mechanical properties, weather resistance, flame retardancy, anti-corrosion and other properties of wood [7,8,9]. However, formaldehyde, acetic acid and other volatile organic compounds (VOCs) are released, which can contaminate the environment and human health.

Silica sol is a colloidal solution with low viscosity and small particle size, which can effectively penetrate into the pore structure of wood and secure a stable three-dimensional network structure with high hardness at the deposited places through gel and drying [10]. Due to its safety and environmental protection, lack of free formaldehyde, good thermal stability and good biocompatibility, silica sol is a promising wood modifier [11]. In the past ten years, many studies have demonstrated that the density, bending strength and flame-retardant properties of wood modified by silica sol have been significantly improved [12]. Silica hydroxy group in silica sol can not only be grafted and copolymerized with other functional groups, but also can be cross-linked with wood cellulose hydroxyl group. It can be used for functional modification of wood cell walls. Silicon is a flame-retardant element, which is green and environmentally friendly. Silica-sol-modified wood meets the requirements of green development. Silica particles can diffuse into the cell wall of wood after being chemically modified by functional reagents (such as alkyl silane, boric acid and titanium dioxide), and the modification effect has been significantly improved [13,14,15,16]. Silica sol and boric acid undergo silanol group condensation to form Si–O–B linkages [17]. Few recent studies have focused on the flame-retardant properties and thermal decomposition kinetics of wood impregnated with alkyl silane and boric-acid-modified silica sol.

In this study, silica sol and boric-acid-modified silica sol were utilized to treat wood and the flame retardation and thermal decomposition kinetics of the modified wood were analyzed. The flame-retardant mechanism of the wood impregnated with boric-acid-modified silica sol was revealed, which can serve as a theoretical basis for fire-retarding treatment of wood.

## 2. Materials and Methods

### 2.1. Materials

Chinese poplar (*Populus tomentosa Carr*) plantation wood (with air-dried density of 0.39 g/cm^3^) was collected from Heilongjiang province in China. Its moisture content was modulated to about 12%. The magnitude of the samples was 300 mm (L) × 25 mm (R) × 25 mm (T). The reagents were purchased from Shanghai Aladdin Bio-Chem Technology Co. Ltd. (Shanghai, China), including tetraethyl orthosilicate (TEOS, analytical grade, CAS 78-10-4), methacryloxy propyl trimethoxy silane melamine (KH570, analytical grade, CAS 2530-85-0) and boric acid (analytical grade, CAS 10043-35-3). The ethanol (≥98.5%, CAS 64-17-5) and hydrochloric acid (HCl, 38%, CAS 7647-01-0) were provided by Beijing Chemical Plant (Beijing, China). The distilled water (H_2_O) was provided by Guangzhou Watson’s Food & Beverage Co., Ltd. (Guangzhou, China). The mole ratio of TEOS/ethanol/H_2_O to synthetic silica sol was 1/4/8. A pH of 2–3 was maintained by the addition of concentrated HCl and monitoring was by a pH meter. The compound modifier was synthesized at the mole ratio of TEOS/ethanol/H_2_O/KH570/boric acid = 1/4/8/8/0.6/0.1. The pH value was 2–3.

### 2.2. Wood Impregnation Treatment

Wood was impregnated with the prepared silica sol and boric-acid-modified silica sol, respectively. Twenty specimens were treated in each group. The solid content of silica sol and boric-acid-modified silica sol is 23.6% and 26.1%, respectively. First, the wood samples were oven-dried and placed into an impregnation chamber. Second, the chamber was evacuated to −0.09 MPa for 0.5 h. Third, the modifier was introduced into the chamber and the vacuum was removed. Thereafter, a positive pressure of about 1.0 MPa was maintained for 24 h. Then the samples were taken out, air-dried to about 50% moisture content and oven-dried to a constant weight with gradually increasing temperature up to 103 °C.

### 2.3. Scanning Electron Microscope

A sample size of 7 mm (L) × 5 mm (R) × 5 mm (T) was taken from the middle part of the mechanical test samples. The microstructures of the samples were observed through a S3400 scanning electron microscope (SEM, Hitachi, Toyko, Japan) after being sprayed gold and combined with energy dispersive X-ray (EDX) spectrometer to scan the silicon element to analyze its micro-consolidation state and distribution in the wood.

### 2.4. Cone Calorimetric Test

A specimen with dimensions of 100 mm × 100 mm × 10 mm was wrapped in aluminium foil. The heat release rate (HRR), time to ignition (TTI), fire performance index (FPI), total heat release (THR), smoke production and yield of carbon oxide of the samples were measured by a cone calorimeter (Fire Testing Technology Instrument, East Grinstead, UK) under an external heat flux of 50 kW/m^2^ according to ISO5660-2002. The side of the specimens with coatings was horizontally exposed to the radiator. Each sample type was measured three times.

### 2.5. Thermogravimetric Analysis

The thermal properties of the wood samples were measured by thermal analyzer (Q500, TGA, Shelton, CT, USA) under nitrogen flow. The temperature was increased from room temperature (25 °C) to 700 °C at heating rates of 10 °C/min, 20 °C/min, 30 °C/min and 40 °C/min. The weight of the samples was kept within 4–5 mg each time. The kinetic parameters were calculated based on the data obtained by the model free iso-conversional methods. The conversion rate α is defined as α = (W_o_ − W_t_)/(W_o_ − W_f_), where W_o_ is the initial weight of the sample, W_f_ is the final residual weight, and W_t_ is the weight of the pyrolyzed sample at time t. The common iso-conversional method used in this study is the method F-W-O [18] (Equation (1)): (1)$$\mathrm{lg}\beta =\mathrm{lg}\left(-\frac{AE}{R\ln(1-\alpha )}\right)-2.315-0.4567\frac{E}{RT}$$
where α is the conversion rate, R is the gas constant (8.314 J·K^−1^·mol^−1^), A is the pre-exponential factor (S^−1^), E is the apparent activation energy (kJ·mol^−1^), T is the absolute temperature (K) and β is the heating rate [19,20]. Therefore, for a given conversion, linear relationships are observed by plotting in lg(β)−(1/T) at different heating rates, and the E is calculated from the slope of the straight line [21].

## 3. Results and Discussion

### 3.1. Microstructure Analysis

SEM and EDX images of the untreated wood (W), the silica-sol-treated wood (W-Si), and the compound-modifier-treated wood (W-Si/B) are shown in Figure 1. The vessel and fiber of W were hollow. After modification, the solid matter filled in the voids in the treated wood (Figure 1b–e). Through the analysis of the silicon X-ray spectrum in the local area (as shown in Figure 1f,g), the solid matter was solidified silicon compound, and there was also silicon distribution in the cell wall of the vessel and fiber.

### 3.2. Cone Calorimetric Analysis

#### 3.2.1. Time to Ignition and Fire Performance Index

As shown in Table 1, W, W-Si, and W-Si/B were set alight after 9 s, 11 s and 15 s, respectively. Treatment with silica sol exhibited longer TTI, suggesting inhibiting the ignition of wood. This happens because silica is deposited in the cell cavity, which can effectively isolate oxygen, increase the thermal conductivity of the material and reduce the combustibility of wood [22,23]. The compound modifier resulted in a higher TTI and increased difficulty to ignite. The FPI is the ratio of TTI and the maxima of the initial peak intensity of the HRR curve. It is an important index for the fire risk assessment of a material. The higher the FPI, the better the fire resistance of the material. The lower the FPI, the more violent combustion will likely be [24]. Compared with W, the FPI of W-Si and W-Si/B increased up to 48.8% and 139.5%, respectively. It suggested that the silica in the system can reduce fire risk and the compound modifier can exhibit better flame-retardant performance.

#### 3.2.2. Heat Release

As shown in Figure 2b, there are two HRR peaks in each group of specimens during the combustion process, and the first HRR peak (p-HRRI) changes rapidly because of the oxidation of the volatile pyrolysis products accumulated on the wood surface [25]. As the burning progresses, an insulating carbon layer appears on the surface of the wood, which leads to a decrease in HRR (the position of the peak and valley of the curve in the figure). When the insulating carbon layer cracks, the wood under the carbon layer begins to burn violently and flashovers occur. The second peak of HRR appears (the moment when the flame burns most intensely) [26]. As shown in Figure 2b, W shows the first peak at 30 s, with a peak value of 179.4 kW/m^2^ (Table 1) and the second peak at 530 s, with a peak value of 207.5 kW/m^2^. The height of HRR curve of modified wood was reduced overall, and the total heat release (THR) curve decreased and became smooth. Compared to W, W-Si exhibited lower p-HRRI and p-HRRII, the p-HRRII decreased by 20.9%, and there was lower total heat release. This indicates that silica can reduce the heat release, due to the glass-like protective covering in wood formed by the silica [27]. The W-Si/B exhibited obviously lower p-HRRI and p-HRRII, resulting in lower THR. As shown in Table 1, the p-HRRII and THR of W-Si/B decreased by 48.0% and 32.1%, respectively. It can be seen that the combination of silica sol and boric acid has an inhibitory effect on the heat-release performance of wood. 

#### 3.2.3. Smoke Production and Yield of Carbon Oxide

The curve of smoke production rate (SPR) and the total smoke production (TSP) are given in Figure 3. The SPR of W exhibited two main peaks (Figure 3a), which is consistent with the HRR (Figure 2b). Smoke consists of incompletely combusted wood and water vapor [28]. Each group of specimens has two SPR peaks. The SPR peak of the modified material is obviously lower than W, so the TSP of the modified material is also lower than W (Figure 3b). These results illustrate that silica sol could effectively suppress the production of smoking. Compared with pure silica sol, the SPR and TSP values of W-Si/B significantly reduced. The TSP decreased to 0.7046 m^2^ (W-Si/B) from 1.3564 m^2^ (W-Si). Acid-catalyzed carbonization results in less carbon residue after burning of modified wood and lower smoke emission during combustion [29]. This is conducive to the formation of carbon layer playing the role of barrier. Boron can also change the structure of char residue, including the expansion degree. Both of these can keep more decomposition fragments in the condensed phase and reduce smoke production. From the SPR and TSR results, it can also be seen that the smoke suppression of the compound modifier was significantly more efficient than that of silica sol.

#### 3.2.4. Mass Loss and Residual Char

As shown in Figure 4, the mass loss (ML) of the W decreased linearly during the initial 500 s of combustion and residual mass remained approximately 7.4% after testing. Compared to the W, treated wood exhibited a similar ML curve through the combustion process, but the residual mass of the W-Si and W-Si/B was higher than the W, increased to 25.6% and 35.6% respectively. Digital photos of the char residues are shown in Figure 4. It can be seen that the untreated sample (Figure 5a) was burned through, and white ash residue remained after burning. The treated samples (Figure 5b,c) showed obvious char formation after the cone calorimetry test, and the shape of the carbon layer remained relatively complete. This is because SiO_2_ gel can function as heat insulator and oxygen barrier and protect the carbon layer. Modified wood promotes char formation under the dehydration effect of the boron compound in the composite modifier, and at the same time, the Si element and the boron element have a synergistic carbon formation effect that promotes the formation of a carbon layer [30,31].

### 3.3. Thermogravimetric Analysis

The curves of residual mass (RM) and derivative thermogravimetric (DTG) are shown in Figure 6. The three stages of wood pyrolysis can be seen from the curve. The first stage of pyrolysis (30–200 °C) is due to dehydration of free water from the specimens [32]. The three samples had a mass loss of 6%. The second stage of wood pyrolysis increased from 200 °C to 400 °C. In this stage, the DTG curve of W had two peaks (shown in Figure 5b), which, respectively, corresponded to the decomposition of hemicellulose and cellulose [33]. When the temperature exceeded 400 °C, further degradation arose, owing to slow decomposition of lignin [34,35]. Compared to untreated wood, the charcoal yield of modified wood was increased, which can be explained by the acid-catalyzed charring. SiO_2_ gel can function as heat insulation and oxygen barrier, and promote the formation of charcoal. Compared to W-Si, the RM increased to 53.1% (W-Si/B) from 34.5% (W-Si), and the peak values of DTG decreased. It can be seen that silicon and boron had a synergistic carbon formation effect, so the thermal degradation rate decreased.

### 3.4. Kinetic Analysis

The lgβ–(1/T) relationship is illustrated in Figure 7. The activation energy (E_a_) and co-relation factors (R^2^) can be obtained by unary linear regression analysis of the lgβ–(1/T) relationship under different heating rates. As seen in Table 2, the E_a_ values of W are between 144 and 186 kJ/mol (in the conversion range 20% to 80%). As the degree of conversion increased, the E_a_ of the wood first increased and then decreased. The activation energy Ea represents the minimum energy required to give rise to a chemical reaction [4]. Compared with the untreated wood, the E_a_ of the modified wood increased, indicating that the minimum energy required to start the thermal decomposition increased. On the one hand, this may have occurred because silica enters the wood and combines with the wood to increase the minimum energy needed for decomposition. On the other hand, the silica in the wood pores will form a closed-barrier layer and effectively shield the wood components making it inaccessible to the air, thereby preventing its pyrolysis [22,23]. Therefore, the introduction of silicon compounds relieves the pyrolysis of wood and improves the flame retardancy of the wood. Compared to W-Si, the E_a_ (in the conversion range 20 to 50%) of W-Si/B decreased. It can be seen that boric acid can reduce the activation energy, and catalyze the thermal degradation and carbonization of poplar wood.

## 4. Conclusions

In our current research, boric-acid-modified silica sol was used to promote flame-retardant properties of poplar wood. The flame-retardant properties and the thermal-decomposition kinetics of modified wood were studied using cone calorimeter analysis and thermogravimetric analysis. The main conclusions of this study are summarized as follows:
(1)Through the full-cell method, modifiers can fill in the vessel and fiber of poplar wood and can effectively enter and be fixed in the wood.(2)The cone calorimeter analysis shows that the ignition time, second peak of heat release rate, total heat release and mass loss of the W-Si/B are obviously delayed. Composite silicon modification has a positive effect on carbonization.(3)Thermogravimetric analysis found that the residual mass of the modified wood increased, and the thermal degradation rate of W-Si/B was significantly lower than others.(4)The E_a_ of modified wood has increased, and the flame-retardant effect of wood is enhanced. Compared to W-Si, the E_a_ (in the conversion range of 20 to 50%) of W-Si/B decreased, because boric acid catalyzed the thermal degradation and carbonization of poplar wood.

In conclusion, boric-acid-modified silica sol can permeate into wood and form a stable three-dimensional network structure. Moreover, wood treated with boric-acid-modified silica sol showed a significant improvement in flame retardancy, compared with wood treated with common silica sol. Therefore, boric-acid-modified silica sol exhibits superior application prospects in fire-retardant wood-based materials.

## Figures and Tables

**Figure 1 materials-13-04478-f001:**
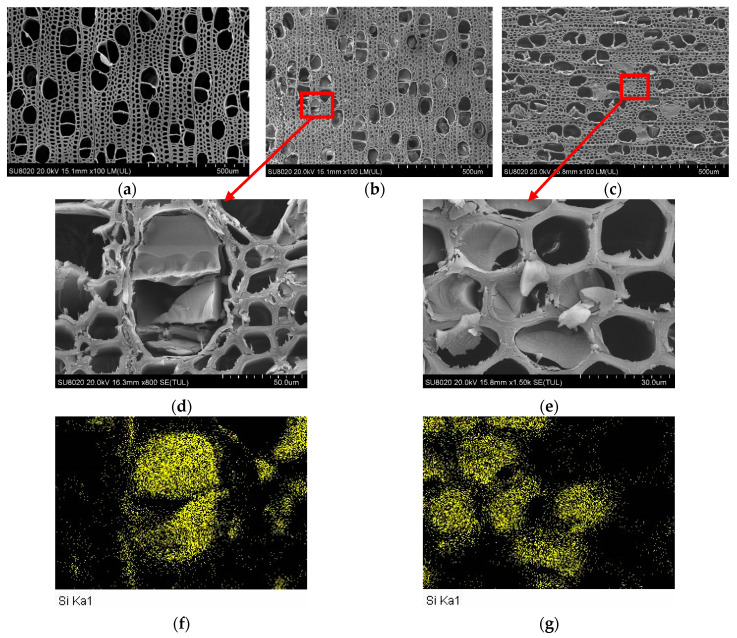
SEM and EDX images of untreated and modified wood. (**a**) cross-section of untreated wood, (**b**) cross-section of W-Si, (**c**) cross-section of W-Si/B, (**d**) magnifying of (**b**), (**e**) magnifying of (**c**), (**f**) silicon X-ray maps of (**d**), (**g**) silicon X-ray maps of (**e**).

**Figure 2 materials-13-04478-f002:**
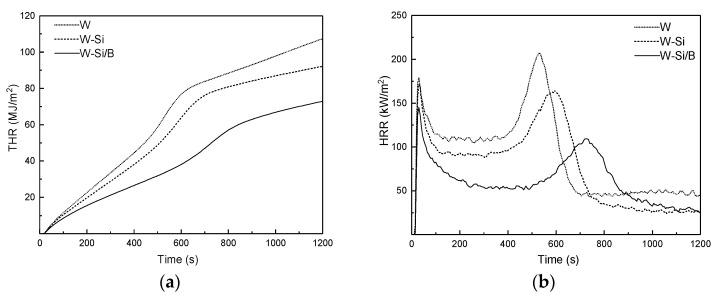
The (**a**) total heat release and (**b**) heat release rate during the cone calorimetric test.

**Figure 3 materials-13-04478-f003:**
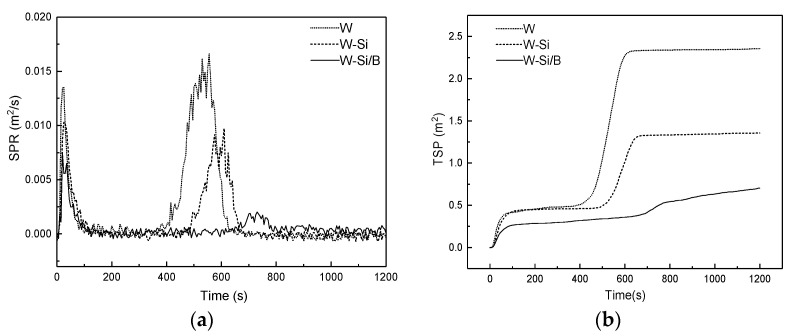
Changes in (**a**) smoke production rate (SPR) and (**b**) total smoke production (TSP) of wood.

**Figure 4 materials-13-04478-f004:**
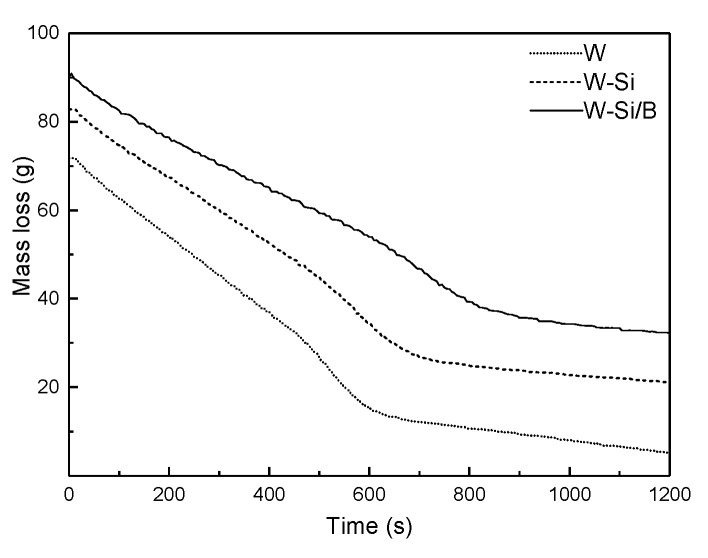
Mass loss of wood during the cone calorimetric test.

**Figure 5 materials-13-04478-f005:**
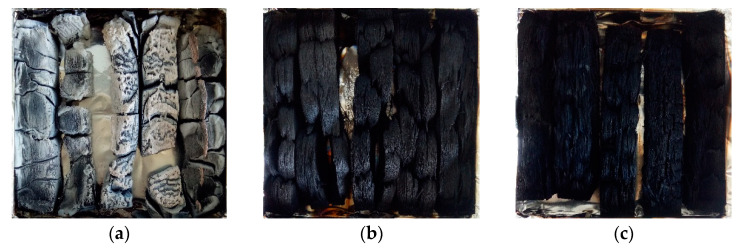
Digital pictures of (**a**) untreated wood, (**b**) W-Si and (**c**) W-Si/B residues.

**Figure 6 materials-13-04478-f006:**
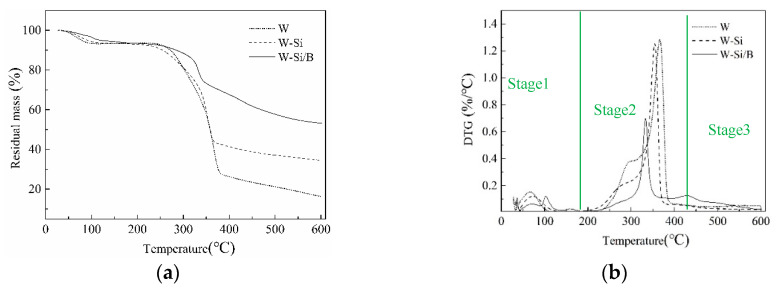
The change in (**a**) residual mass and (**b**) derivative thermogravimetric.

**Figure 7 materials-13-04478-f007:**
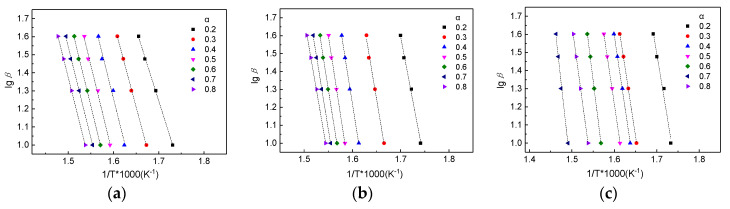
The lgβ-1/T curve of (**a**) untreated wood, (**b**) W-Si and (**c**) W-Si/B residues.

**Table 1 materials-13-04478-t001:** Cone calorimetry test results of samples.

Sample	TTI (s)	FPI (m^2^ s/kW)	p-HRR1 (kW/m^2^)	p-HRR2 (kW/m^2^)	THR (MJ/m^2^)	TSP (m²)
W	9	0.043	179.4	207.5	107.4	2.35
W-Si	11	0.064	171.4	164.2	92.2	1.35
W-Si/B	15	0.103	145.4	109.4	72.9	0.7

**Table 2 materials-13-04478-t002:** Kinetic parameters of samples.

Degree of Conversion α	W		W-Si		W-Si/B	
	E_a_ (kJ/mol)	R^2^	E_a_ (kJ/mol)	R^2^	E_a_(kJ/mol)	R^2^
0.2	144.0	0.9989	257.27	0.9928	258.3	0.9757
0.3	170.8	0.9994	292.16	0.9913	278.3	0.9993
0.4	182.2	0.9786	308.83	0.9986	287.2	0.9992
0.5	186.6	0.9904	319.50	0.9966	286.5	0.9997
0.6	186.4	0.9960	311.13	0.9970	342.9	0.9994
0.7	184.0	0.9981	301.91	0.9987	385.1	0.9865
0.8	181.8	0.9998	282.82	0.9989	314.2	0.9991

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
