# Peer review of "Flame Retardant Properties and Thermal Decomposition Kinetics of Wood Treated with Boric Acid Modified Silica Sol"

_materials, 2020, doi:10.3390/ma13204478_

Round 1

Reviewer 1 Report

The paper reports on a study of the flame retardant properties of wood treated with boric acid modified silica sol. The authors used relevant techniques to characterize the treated wood and the results are interesting. Overall, the work is well designed and performed and can be accepted after some minor corrections as follow: 1 Introduction lines 41-43, “In recent years many studies…..[12]”. The recent studies on the topic should be more clearly presented. 2 Materials and Methods lines 68-73. The impregnation methodology should be presented more precisely including amounts of the treated wood and the impregnating solution.

Author Response

Response to Reviewer 1 Comments

Point 1: Introduction lines 41-43, “In recent years many studies…..[12]”. The recent studies on the topic should be more clearly presented.

Response 1: Thank you for your suggestion. The corresponding research background introduction has been added in the article.

Point 2: Materials and Methods lines 68-73. The impregnation methodology should be presented more precisely including amounts of the treated wood and the impregnating solution.

Response 2: The prepared silica sol and boric acid modified silica sol were used to impregnate the wood, 20 specimens in each group. The solid content of silica sol and boric acid modified silica sol is 23.6% and 26.1%, respectively. It has been added to the text.

Reviewer 2 Report

Dear Authors, I find the article well written and containing valuable results, but as usual there are some suggestions to improve it.

Line  68 - Wood impregnation treatment. Question: How much of the modifier is contained in the wood samples? Was it estimated?

Line 189 - "As saw in Table 2, the Ea values of W are between 144 and 186 kJ/mol". Discussion: In this article kinetics is treated formally and its connection to the chemical decomposition is not clear, especially in relation to the calculated numeric values of activation energy in kJ/mol. Is it possible in the case of wood samples to express the quantity in moles? What is the mole of wood related to?

Table 2

Accuracy for Ea to high, one decimal place is enough

Line 197

Therefore, the introduction of silicon compounds relieves the pyrolysis of wood and improves the flame retardants of the wood.

Question: What do you understand by "relieves the pyrolysis"?

Instead of "improves the flame retardants" (is there any interaction?) I propose to write "improves the flame retardancy"

Line 200:  " reduce the activation energy, and catalyzed the thermal degradation " CORRECT : reduce the activation energy and  catalyze the thermal degradation

Line 207: were studied with cone calorimetric analysis CORRECT were studied using cone calorimeter (there is no cone calorimetry)

Line 217: was decreased, which because boric acid catalyzed CORRECT  was decreased, because boric acid catalyzed

Author Response

Response to Reviewer 2 Comments

Point 1: Line68 - Wood impregnation treatment. Question: How much of the modifier is contained in the wood samples? Was it estimated?

Response 1: The weight percent gain of W-Si and W-Si/B are 28.3% and 59.1%, respectively. Calculated according to the increase rate of absolute dry mass of wood after impregnation

Point 2: Line189 - "As saw in Table 2, the Ea values of W are between 144 and 186 kJ/mol". Discussion: In this article kinetics is treated formally and its connection to the chemical decomposition is not clear, especially in relation to the calculated numeric values of activation energy in kJ/mol. Is it possible in the case of wood samples to express the quantity in moles? What is the mole of wood related to?

Response 2: Pyrolysis kinetics is the analysis and study of the thermochemical conversion mode during the thermal decomposition of biomass materials. The magnitude of activation energy represents the minimum energy required for chemical reactions. With reference to the literature of previous studies, we selected a kinetic formula suitable for the wood pyrolysis mode to calculate the activation energy. Wood is a natural organic material with complex composition and it is difficult to express in mole.

Point 3: Table 2Accuracy for Ea to high, one decimal place is enough.

Response 2: Modified in the original text as suggested.

Point 4: Line 197 Therefore, the introduction of silicon compounds relieves the pyrolysis of wood and improves the flame retardants(reterdancy)of the wood. Question: What do you understand by "relieves the pyrolysis"?

Response 2: "Relieving pyrolysis" means that silicide makes it difficult for wood to undergo thermal decomposition.

Point 5: Instead of "improves the flame retardants" (is there any interaction?) I propose to write "improves the flame retardancy"

Response 5: Modified in the original text according to your suggestions.

Point 6: Line 200: " reduce the activation energy, and catalyzed the thermal degradation "

 CORRECT : reduce the activation energy and catalyze the thermal degradation

Line 207: were studied with cone calorimetric analysis

CORRECT were studied using cone calorimeter (there is no cone calorimetry)

Line 217: was decreased, which because boric acid catalyzed

CORRECT was decreased, because boric acid catalyzed

Response 6: Has been changed in the text according to the above suggestions.

Reviewer 3 Report

Review for manuscript: materials-945139

Title: Flame retardant properties and thermal decomposition kinetics of wood treated with boric acid modified silica sol

Authors: Qiangqiang Liu, Lin Ni *, Yubo Chai, Wenhua Lyu Submitted to section: Biomaterials, https://www.mdpi.com/journal/materials/sections/biomaterials

The paper addresses experimental research on flame retardant properties and thermal decomposition kinetics of wood treated by boric acid modified silica sol. In general the paper content is relevant from the scientific point of view and addresses important issues relevant for the fire safety engineering community and building construction sector, considering the wood behaviour and protection against fire.

The paper is recommended for publication with some minor revisions and suggestions.

In section 2.5 the FWO method was used with a linear reaction order. Further background is need for this assumption, and its implications on the calculation of the apparent activation energy.

A reference for the FWO method and eq 1 is need. (Line 95)

In section 3.2.1, line 118, the introduction on the fire performance index (FPI) is made. More information about this index is need, explaining the importance to wood fire protection.

In line 130 the term flashover is used to describe the combustion/carbonization of bottom layers. It seems inadequate.

Figure 6 (b), in line 173, presents the DTG curves. For the W-Si/B sample two small peaks are shown at 100ºc and between 400-450ºc. A more detailed information about them could be given, comparing the difference to the other curves.

Figure 7, line 202, show the lgβ-1/T curves from the conversion ranges 20-80%. An explanation must be given for this range, and the relation with the applicability of eq. 1.

Author Response

Response to Reviewer 3 Comments

Point 1: In section 2.5 the FWO method was used with a linear reaction order. Further background is need for this assumption, and its implications on the calculation of the apparent activation energy. A reference for the FWO method and eq 1 is need. (Line 95)

Response 1: The FWO method is a common method for calculating the activation energy Ea. It is used by reading literature and then referring to it. For details, see References.(Ozawa T. A New Method of Analyzing Thermogravimetric Data[J]. Bull Chem Soc Jpn, 1965, 38(11): 1881-1886.)

Point 2: In section 3.2.1, line 118, the introduction on the fire performance index (FPI) is made. More information about this index is need, explaining the importance to wood fire protection.

Response 2: The Fire Performance Index (FPI) is related to the flashover of wood. The larger the Fire Performance Index (FPI), the later the time for the wood to crash and the time for the wood products to collapse. It has been added to the text.

Point 3: In line 130 the term flashover is used to describe the combustion/carbonization of bottom layers. It seems inadequate.

Response 3: The word originally intent to express that the wood burns on the surface first and then burns completely inside. Original sentence was revised.

Point 4: Figure 6 (b), in line 173, presents the DTG curves. For the W-Si/B sample two small peaks are shown at 100ºc and between 400-450ºc. A more detailed information about them could be given, comparing the difference to the other curves.

Response 4: The small peak at 100ºC may be due to the delayed evaporation of water in the wood. The 400ºC may be due to the increase in temperature during the char formation process, which causes the undegraded part covered by the char to continue to degrade. Since the loss of these two parts is very small, it has little effect on the main degradation stage of wood, so detailed research is not carried out.

Point 5: Figure 7, line 202, show the lgβ-1/T curves from the conversion ranges 20-80%. An explanation must be given for this range, and the relation with the applicability of eq. 1.

Response 5: The conversion rate of 20-80% is selected based on literature research, and the literature describes that the linear correlation within this range is better. At the same time, because 20-80% is also the main stage of wood thermal degradation (most of the quality loss), it is the focus of our analysis. The relevant documents are as follows.(Yao F, Wu Q, Lei Y, et al. Thermal decomposition kinetics of natural fibers : Activation energy with dynamic thermogravimetric analysis[J]. Polymer Degradation and Stability, 2008, 93(1): 90-98.)
